# Pastoralism in the highest peaks: Role of the traditional grazing systems in maintaining biodiversity and ecosystem function in the alpine Himalaya

Tenzing Ingty 🔟 *

Department of Biology, University of Massachusetts, Boston, Massachusetts, United States of America

* tenzingingty@gmail.com

**Data Availability Statement:** All relevant data are within the manuscript and its Supporting information files.

**Funding:** Field work was supported by funding from 1. Department of Biotechnology, Ministry of

## Abstract

Rangelands cover around half of the planet's land mass and provide vital ecosystem services to over a quarter of humanity. The Himalayan rangelands, part of a global biodiversity hotspot is among the most threatened regions in the world. In rangelands of many developing nations policies banning grazing in protected areas is common practice. In 1998, the Indian state of Sikkim, in the Eastern Himalaya, enacted a grazing ban in response to growing anthropogenic pressure in pastures and forests that was presumably leading to degradation of biodiversity. Studies from the region demonstrate the grazing ban has had some beneficial results in the form of increased carbon stocks and regeneration of some species of conservation value but the ban also resulted in negative outcomes such as reduced household incomes, increase in monocultures in lowlands, decreased manure production in a state that exclusively practices organic farming, spread of gregarious species, and a perceived increase in human wildlife conflict. This paper explores the impact of the traditional pastoral system on high elevation plant species in Lachen valley, one of the few regions of Sikkim where the grazing ban was not implemented. Experimental plots were laid in along an elevation gradient in grazed and ungrazed areas. Ungrazed areas are part of pastures that have been fenced off (preventing grazing) for over a decade and used by the locals for hay formation. I quantified plant species diversity (Species richness, Shannon index, Simpson diversity index, and Pielou evenness index) and ecosystem function (above ground net primary productivity ANPP). The difference method using movable exlosure cages was used in grazing areas to account for plant ANPP eaten and regrowth between grazing periods). The results demonstrate that grazing significantly contributes to greater plant species diversity (Species richness, Shannon index, Simpson diversity index, and Pielou evenness index) and ecosystem function (using above ground net primary productivity as an indicator). The multidimensional scaling and ANOSIM (Analysis of Similarities) pointed to significant differences in plant species assemblages in grazed and ungrazed areas. Further, ecosystem function is controlled by grazing, rainfall and elevation. Thus, the traditional transhumant pastoral system may enhance biodiversity and ecosystem function. I argue that a complete restriction of open grazing meet neither conservation nor socioeconomic

Science and Technology Government of India to Ashoka Trust for Research in Ecology and the Environment, Bangalore, India (atree.org) 2. ATREE Small Grants for research in Northeast India to TI 3. Sanofi Genzyme and Oracle Fellowship to TI The funders had no role in study design, data collection and analysis, decision to publish, or preparation of the manuscript.

**Competing interests:** TI received graduate student scholarship from Sanofi Genzyme and Oracle fellowship (a commercial source). This does not alter our adherence to PLOS ONE policies on sharing data and materials.

goals. Evidence based policies are required to conserve the rich and vulnerable biodiversity of the region.

## Introduction

Rangelands cover approximately half of earth's land surface [1,2] and over a quarter of humanity depend on these land types for their livelihoods and numerous ecosystem services [3,4]. The ecosystem services include provisioning services like food, fiber and water, and supporting services such as carbon storage (roughly 30% of global soil carbon) among many others [5–8]. Rangelands are also recognized for their high biodiversity, including numerous endangered and endemic species, as well as habitat connectivity between protected forests [6,7,9,10]. In much of the developing world, rangelands are inhabited by pastoral communities. Pastoralism that has existed for millennia has impacted the evolution of many rangelands, from the biodiverse rich African savanna [11] to the highest elevations in the Tibetan steppe [12]. Pastoralists and the rangelands they inhabit thus form complex socio-ecological systems that have co-evolved across different geographies, biodiversity and climates [11,13]. Many pastoralists have been blamed, often wrongly, for rangeland degradation [14,15] and targeted for "*modernization*" programs such as sedenterization. While some of these programs have been beneficial [6,16–18], most have shown to have socially and ecologically disastrous impacts resulting in a decline in wildlife, decreased grassland diversity and productivity, and increased desertification [14,19,20]. Moreover, the abandonment of rangelands by pastoralists and climate change is leading to rapid shrub invasion, further increasing habitat loss [21,22]. Grazing by herbivores such as the semi domesticated reindeer has been shown to counteract the effect of warming on shrub expansion [23–26]. This suggests that grazing management is an important factor to maintain alpine areas in response to climate change driven shrub invasion.

Rangelands in the Himalaya, a global biodiversity hotspot [27], form the largest land use system and provides numerous ecosystem services to pastoral communities [28]. The Himalaya is the source of 10 of the largest rivers in Asia whose basins feed about one fifth of humanity. The alpine rangelands of the Himalaya thus play a vital role in retaining critical ecosystem services such as carbon sequestration, water storage and provisioning, maintaining biodiversity, and food security and livelihoods. Despite its importance a report on high elevation rangelands of the Himalaya [28] underline numerous threats such as invasive species, inappropriate management and development policies, overgrazing and climate change. Studies show that climate change has significantly impacted the Himalaya. These include an increase in average temperatures three times higher than the global average, altered phenology and shifts in distribution ranges of biodiversity [29–33]. Upward shifts in treelines and shrubs have led to woody ingression into alpine pastures [34–36].

The alpine rangelands of the tiny state of Sikkim in the Eastern Himalaya are among the most biodiverse regions in the world. They occupy just 0.006% of global alpine rangelands but harbor as much as 60% of the global alpine plant families and 10% (297 genera) of global alpine genera with more than twice the number of vascular plants than the Alps and Rockies [37]. In 1998 the government of Sikkim adopted a grazing exclusion policy in response to growing anthropogenic pressure on grazing lands and protected forests leading to perceived degradation. Open grazing was phased out over the next six years in all reserved forests with exception of alpine pastures in the Kangchenzonga National Park [38,39], Lachen and Lachung valleys [40]. The forest cover of the state between 1999 to 2017 increased by about 4% from 3118 km$^2$ to 3379 km$^2$. Studies in Barsey Rhododendron Sanctuary in West Sikkim estimate that twelve years after the declaration of the grazing ban carbon stocks increased by about 585 thousand

tonnes and the concurrent increase in regeneration of species of high conservation value such as *Rhododendron barbatum* [41,42]. At the same time, Bhagwat et al. [41] also warned of insignificant regeneration of important fodder species and other species of high conservation value such as *Quercus spp.* coupled with a spread of gregarious species such as *Arundinaria maling* and *Viburum cordifolium*. Restriction on free access to forest resources have had significant socio-economic impacts on the herders who were dependent on grazing pastures in the sanctuary. Their average net income from livestock was reduced by more than half the pre-ban period, with the source of majority of the incomes now coming from government subsidies and schemes like the National Rural Employment Guarantee Act programme (MGNREGA) for unskilled labor [41,43]. A study on the perception of sedenterized ex-herders from West Sikkim identified loss of traditional livelihoods leading to decreased income and limited land holdings as their main protestations. In addition to socio-economic impacts to herders the grazing ban has had a cascading impact on non-herders [41]. Perceptions of local villagers indicate an increase in human wildlife conflict in the form of crop raids and livestock depredation with over 85% of the households experiencing greater losses in agricultural outputs after the grazing ban [41,43].

Here, I explore the impact of the present pastoral system on high elevation plant species in Lachen valley, Sikkim. The region is of significance for two reasons. First, it falls in one of the most biodiverse regions of the world, the alpine zone of the Sikkim Himalaya [37]. Second transhumant pastoralism has been practiced in the region for centuries with the indigenous communities directly dependent on the biodiversity of their surrounding for livestock husbandry, water storage and provisioning, and the collection of medicinal, aromatic, and edible plants [38,44,45].

I address two broad questions:

1. What is the impact of the traditional grazing system on plant diversity and ecosystem function (using aboveground net primary productivity as an integrative measure of ecosystem function [46,47]) in the high-elevation alpine Himalaya?

2. How does grazing maintain biodiversity and ecosystem function in the region?

I first analyze the impact of grazing on diversity and productivity. I then discuss the role of grazing in maintaining high biodiversity and ecosystem function. Finally, I conclude by identifying further knowledge needs and propose recommendations for an improved grazing policy to protect the rich yet vulnerable biodiversity of the region.

## Materials and methods

### Study area

The study was conducted in Lachen valley in the North district of the state of Sikkim in India. Necessary field research permits were obtained from the Department of Forest, Environment & Wildlife Management, Government of Sikkim, India—Letter No. 16/SRO/F dated 20/04/2010 and Ref. No. 101 /PCCF/F dated 03/02/2012. Sikkim, in the Eastern Himalayan region (Fig 1), is the second smallest (7096 km2) and the least populous state of India. The alpine zone of Sikkim has unique biodiversity since it also falls in the transition zone of two biogeographic regions, namely the central Himalaya region and the southern fringe of the Palearctic region [48]. Lachen valley lies within the alpine zone of Sikkim and shares an international border with the Tibet Autonomous Region (TAR) of China. The valley is among the few places where open grazing is permitted in the state of Sikkim [40]. It is inhabited by two pastoral communities—the yak and sheep herders, the Dokpas and the agro-pastoral Lachenpas.

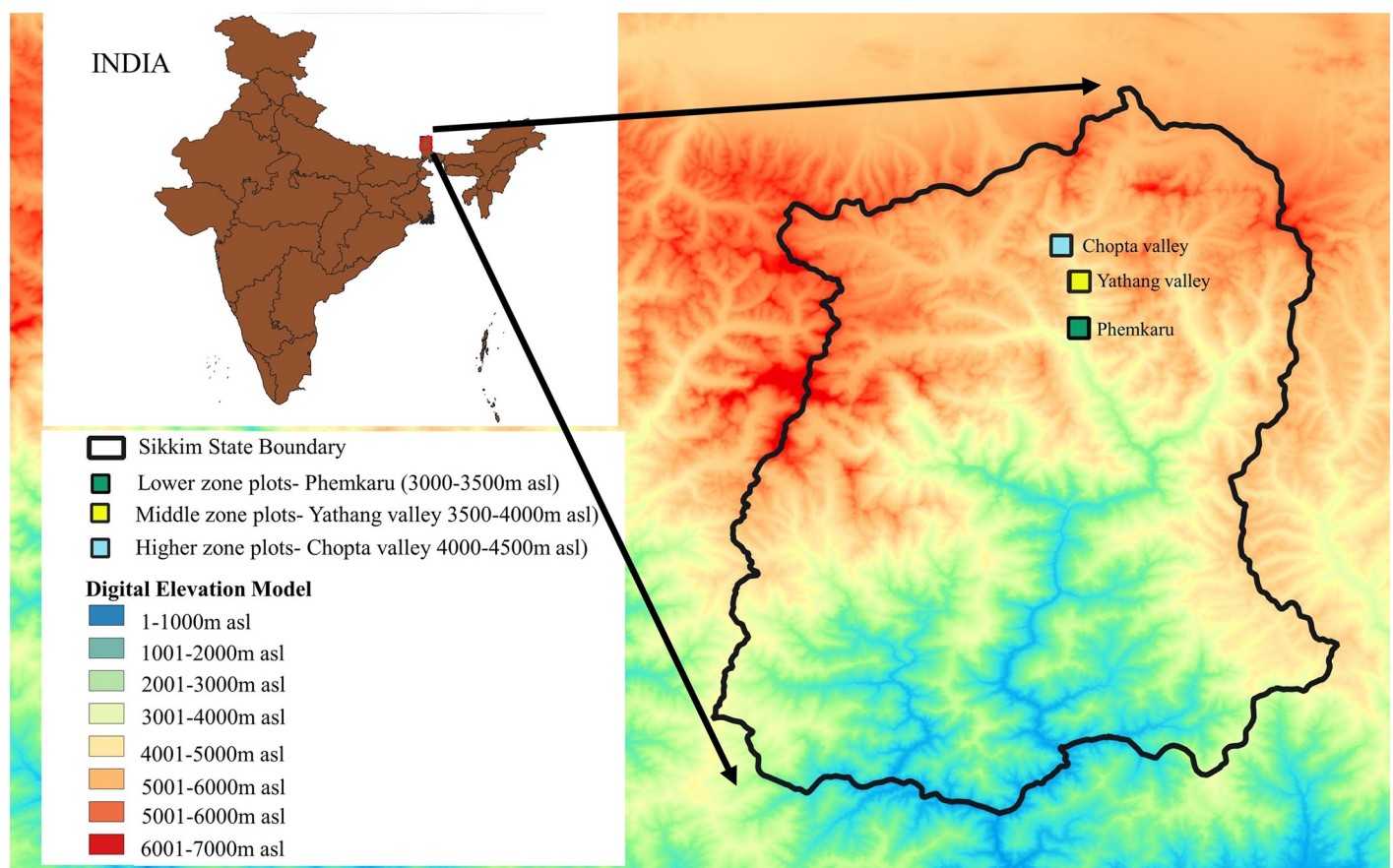

**Fig 1. Map of study site with experimental plots.** Elevation classes displayed were prepared using Digital Elevation Model (DEM) derived from ASTER data [52].

According to the states 2012 livestock census the communities rear 951 heads of cattle, 2588 yaks, 141 horses, 230 sheep (indigenous variety) and 90 goats [49]. Historically both communities conducted trade and practiced pastoralism [50,51]. They used to take their yak and sheep upto Khambazong in TAR, China in summer and traded oil, sugar and fir planks for wool, barley flour (*tsampa*), salt, carpets, mutton and sheep fat. After the 1962 Sino-Indian war the international border was closed off. Now, both communities practice seasonal transhumance. The Lachenpas move to the high elevation pastures in the summer and retreat to lower elevations in the winter, while the Dokpas do the opposite, moving onto the Tibetan Plateau where they find wind-blown, snow-free pastures in winter. Both communities rely on common property resources (pastures, forests, and natural waters) and some pastures on state forest department and military property.

## Vegetation sampling

Herb and graminoid species were sampled in grazing pastures and ungrazed areas. Grazed areas were defined as pastures most often in use and that have been grazed for atleast two decades. The ungrazed areas were defined as areas with a history of no grazing for ateast 10 years with permanent fences around them. These areas are used by the locals to make hay. The herbs s are left to grow in these areas throughout the year and removed at the end of the growing season. Local pastoralists and village representatives were consulted to select grazed and

ungrazed areas. I used a randomized block design with two treatments (grazing and ungrazed) and three replicates (lower zone = 3000-3500m; middle zone = 3500-400m; and higher zone = 4000-4500m) making a total of six 30mX 30m plots (2 treatments X 3 replicates). All experimental data was collected over a two-year period starting from the growing season of 2013 to the end of the growing season in 2015. It was difficult to separate the effects of different grazers since there was much overlap in the grazing areas shared by the different livestock and wild species. Cows, bulls, goats, horses and a few yak shared the same pastures in the lower zone (3000–3500 m asl). Sheep, yak, cows, bulls, goats, and horses shared the middle zones (3500–4000 m asl) and yak, sheep and a few horses grazed in the higher zone (4000–4500 m asl). Thus, I studied the combined effects of different grazing species.

## Impacts of grazing on species diversity and above ground net primary productivity (ANPP)

**Diversity.** In each 30mX30 m plot 12 quadrats (1mX1 m) were randomly laid. I had 36 quadrates in grazed areas (12X 3elevation zones) and 36 in ungrazed areas. Thus, a total of 72 quadrats were laid (24 in each zone/ replicate). Data on species composition and percentage cover was collected from each quadrat for herb species only (forbs and graminoids). The grazing pastures I sampled had a negligible amount of woody vegetation and none fell in our quadrats. In the ungrazed plots there were few shrubs, only along the edges which I anyway exluded to avoid for edge effect.

**ANPP.** I followed the difference method by McNaughton et al. [53] to calculate ANPP in the grazed area. Twelve temporary movable exclosure cages, 1mX1m in size, were set up in each grazed plot. Thus a total of 36 movable exclosure cages were setup every month (12 cages in 3 elevation zones). After each sampling effort (monthly) these cages were moved to a new location within the grazed plot to account for biomass growth (over a month). The above ground standing biomass (vegetation) was clipped to ground level and collected inside and outside exclosure cages. The collected plant material was oven dried 24 hours at 60°C and then weighed. The ANPP of grazing plots was calculated using the following formula [53]

$$ANPP = W1g + (W2u - W1g) + (W3u - W2g) + (W4u - W3g) + (W5u - W4g)$$

where Wi is the dry mass weight of above ground standing biomass at sample time ti (i = 1, 2, 3, 4, 5: beginning of June, July, August, September, October respectively), u (ungrazed) and g (grazed) represent samplings inside and outside the exclosure cages, respectively. In ungrazed areas twelve 1mX1m quadrats were laid and biomass was collected at peak standing crop (annually).

## Data analysis

Diversity indices—species richness (S), shannon index (H), simpson diversity index (D) and pielous's evenness (J) were calculated for plant species in grazed and ungrazed quadrats in each elevation zone and for all quadrats (all elevations and grazing treatments combined). Proportional dominance (PD) of a species was calculated as a function of relative density and frequency (cover). $PD = \left(\frac{Ni}{N}\right)*fi$ where PD is the proportional dominance of species i, Ni is the percent cover of species i in all locations, N is the percent cover of all species in all locations, and fi is the frequency of plots at which species i occurs. The Kruskal-Wallis test was used to calculate significance for difference between diversity indices and ANPP in grazed and ungrazed areas and different elevation zones.

To compare the differences between assemblages in terms of rarity (Leroy's rarity index [54,55]), heterogeneity of the grassland community (Menhinick index [56]), evenness

(McIntosh index [57])and taxonomic distinctness [58,59] (Dstar) the four components of diversity were combined into a single measurement using the DER_algorithm in the EcoIndR package in R [60]. The algorithm calculates the polar coordinates of all samples with all possible combinations for the four indices and then calculates the convex hull and mean Euclidean distances between samples. Each circle represents a quadrat (1mX1m), the size of the circles represents species richness and the color rarity.

To explore the response of species composition to grazing treatments and elevation a triangular similarity matrix was constructed using Bray Curtis similarity coefficients based on cover data. The similarity matrix was displayed in an ordination plot using non-metric multi-dimensional scaling (NMDS). The stress function assesses how well the calculated sample relationships are represented in the two-dimensional plot. A lower value (<0.2 commonly accepted) represents a more accurate ordination [61,62].

The Analysis of similarity (ANOSIM) was used to check the degree of dissimilarity of species across grazing treatments, elevation zones, and treatments within each elevation zone. ANOSIM based on the Bray Curtis similarity matrix was used to check significance levels using a permutation test with 999 simulations. R values close to 1 suggest sites within a group are similar to each other and dissimilar to sites in other groups. While an R value close to "0" suggests an even distribution of high and low ranks within and between groups. A linear regression was used to examine the relationship between diversity and productivity.

All analyses were performed on R. The NMDS and ANOSIM analysis were performed using the vegan package in R [63].

## Results

### Species diversity indices

A total of 153 plant species between 3000–4500 m asl were sampled. This included the 66 (S = 13.38 ±5.71),56 (S = 11.42 ±5.01) and 47 (S = 9.38 ±3.41) species found in the lower, middle, and higher zones, respectively. A significantly (p<0.001) higher number of species was found in grazed areas (114 species) than in ungrazed areas (61 species). I sampled 51,48 and 30 species in grazed areas and 22,19 and 22 species in ungrazed areas in lower, middle, and higher zones, respectively. The analysis showed all diversity indices (S,H,D and J) were significantly higher (p<0.001) in grazed quadrats than ungrazed quadrats in all elevation zones combined and in each zone (Fig 2A–2D, Table 1). Grazing had a significant (p<0.001) effect on all diversity indices in each elevation zone and all elevation zones combined (Table 1). Unlike grazing, elevation did not have a significant effect on diversity indices (Fig 2F–2I).

Results of the ANOSIM indicate statistically significant differences in species composition. There is a significant difference in species composition between grazing treatments in each elevation zone (lower: R = 0.0.911 p<0.001;middle: R = 0.786 p<0.001; higher R = 0.878 p<0.001) (Fig 3C–3E). As would be expected, elevation plays a significant role in species composition. Species composition (grazed and ungrazed combined) in different elevation zones showed a significant difference (R = 0.844 p<0.001) (Fig 3B). The ANOSIM R value for grazing treatment across all elevations was relatively lower (R = 0.199 p<0.001) suggesting comparatively more overlap in species compositions in grazed and ungrazed areas (Fig 3). Overall, grazed and ungrazed plots had 14.38% (22) species in common. In the lower, middle and higher zones they shared 7(10.61%), 11 (19.64%) and 5 (10.64%) species, respectively.

In addition to S,H,D and J, grazing also affected the other components of diversity (Fig 4). Fig 4 shows high dispersion between grazed quadrats and ungrazed quadrats. Grazed quadrat number 5 in the lower zone (G5) showed the highest values of rarity, heterogeneity, evenness and taxonomic distinctness whereas U14 –ungrazed quadrat number 14 in the middle zone

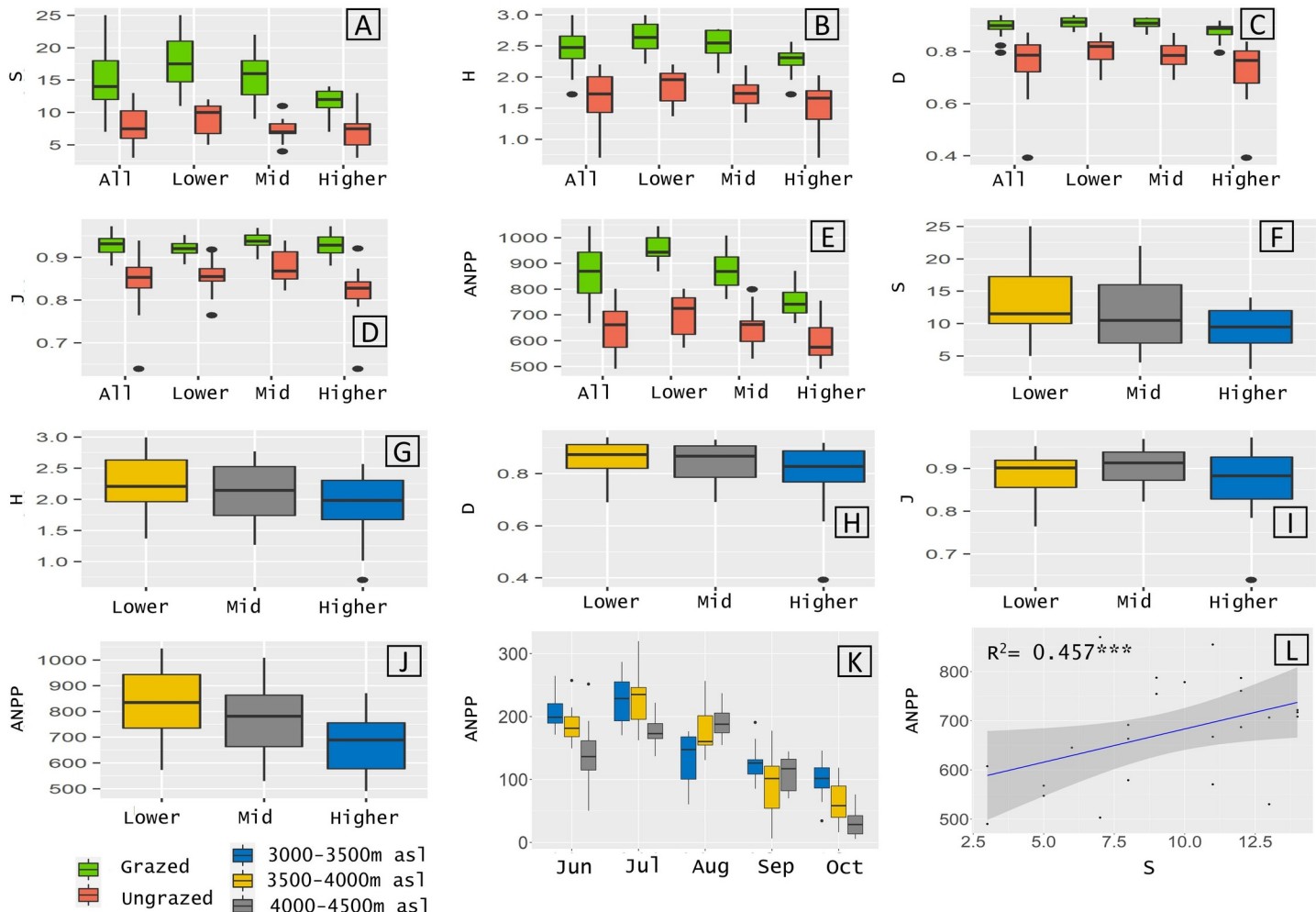

**Fig 2.** A-J Boxplots of diversity indices and above ground net primary productivity (ANPP) for 153 species sampled from 72 plots. A-E diversity indices and ANPP in grazed vs ungrazed plot in all elevations combined (All) and in each elevation zone (lower, middle and higher). F-J diversity indices and ANPP in each elevation zone for all species. K -ANPP in each elevation zone per month (June to October). L -Linear relationship between species richness (quadrate scale) and ANPP in lower, middle and higher elevation zone quadrates. $R^2$ values indicated. *** represents statistical significance (p<0.01).

showed the lowest values for all indices (Fig 4). As is evidenced from the size of the circle G5 has markedly higher species richness than U14. The color and size of the circles indicate species richness and rarity among grazed plots compared to ungrazed plots. Rarity and richness was higher in grazed quadrats (G) than ungrazed quadrats (Fig 4).

## Species proportional dominance

All plots were dominated by Graminoid species (Cyperaceae, Juncaceae and Poaceae). They represented 70.58%, 44.34% and 78.19% of PD in all quadrats combined, grazed quadrats and ungrazed quadrats, respectively. These included sixteen species (Cyperaceae (n = 7), Juncaceae (n = 2(, Poaceae (n = 7)) in total. In grazed quadrats I found thirteen graminoid species (Cyperaceae (n = 6), Juncaceae (n = 2), Poaceae (n = 5)) and ten graminoid species (Cyperaceae (n = 4), Juncaceae (n = 1), Poaceae (n = 5)) in ungrazed quadrats. Forbs showed high PD in grazed quadrats than ungrazed quadrats with Polygonaceae, Primulaceae, Rosaceae, Asteraceace and Orchidaceae families representing 37.75% of PD.

**Table 1. Diversity indices and above ground net primary productivity (average per quadrate ±standard deviation) for plots in grazed and ungrazed areas in the three elevation zones and all elevations combined.**

| Elevation Zone | Quadrates | S | H | D | J | ANPP (g/m$^2$) |
|---|---|---|---|---|---|---|
| All elevations combined | All | 11.39 ±5.01 | 2.08 ±0.5 | 0.83 ±0.09 | 0.89 ±0.06 | 756.46 ± 143.33 |
| | Grazed (G) | 14.92 ±4.3 | 2.47 ±0.28 | 0.9 ±0.03 | 0.93 ±0.02 | 861 ±105.63 |
| | Ungrazed (U) | 7.86 ±2.61 | 1.70 ±0.35 | 0.77 ±0.09 | 0.84 ±0.05 | 651.92 ±89.65 |
| | Difference (G-U) | **7.06**\*\*\* | **0.77**\*\*\* | **0.13**\*\*\* | **0.08**\*\*\* | **209.08**\*\*\* |
| Lower elevations | All | 13.38 ±5.71 | 2.23 ±0.49 | 0.85 ±0.07 | 0.89 ±0.05 | 829.77 ±144.73 |
| | Grazed (G) | 17.83 ±4.3 | 2.63 ±0.26 | 0.91 ±0.02 | 0.92 ±0.02 | 955.27 ±52.24 |
| | Ungrazed (U) | 8.92 ±2.5 | 1.84 ±0.3 | 0.80 ±0.58 | 0.86 ±0.04 | 704.26 ± 81.87 |
| | Difference (G-U) | **8.91**\*\*\* | **0.79**\*\*\* | **0.11**\*\*\* | **0.06**\*\*\* | **251.01**\*\*\* |
| Middle elevations | All | 11.42 ±5.01 | 2.13 ±0.48 | 0.85 ±0.07 | 0.90 ±0.04 | 764.68 ±135.45 |
| | Grazed (G) | 15.33 ±3.8 | 2.53 ±0.24 | 0.90 ±0.02 | 0.94 ±0.03 | 873.76 ±79.28 |
| | Ungrazed (U) | 7.50 ±2.11 | 1.73 ±0.28 | 0.79 ±0.06 | 0.88 ±0.04 | 655.60 ±78.18 |
| | Difference (G-U) | **7.83**\*\*\* | **0.8**\*\*\* | **0.11**\*\*\* | **0.06**\*\*\* | **218.16**\*\*\* |
| Higher elevations | All | 9.38 ±3.41 | 1.90 ±0.49 | 0.80 ±0.12 | 0.87 ±0.08 | 674.93 ±107.29 |
| | Grazed (G) | 11.58 ±2.15 | 2.26 ±0.23 | 0.88 ±0.04 | 0.93 ±0.03 | 753.96 ±64.13 |
| | Ungrazed (U) | 7.17 ±3.01 | 1.54 ±0.41 | 0.72 ±0.13 | 0.82 ±0.07 | 595.90 ±79.56 |
| | Difference (G-U) | **4.41**\*\*\* | **0.72**\*\*\* | **0.16**\*\*\* | **0.11**\*\*\* | **158.06**\*\*\* |

P-values (represented by asterixis) indicate difference between grazed and ungrazed areas based on the Kruskal-Wallis. S = Species richness; H = Shannon index; D = Simpsons diversity index; J = Pielou's evenness; ANPP = Above ground net primary productivity.

P values- <0.001 "\*\*\*" <0.01 '\*\*' <0.05 '\*'.

At the species level *Kobresia nepalensis* (Cyperaceae) was the most dominant species in grazed and ungrazed plots. The five most dominant species in all quadrats combined represented 55.95% of PD. In grazed and ungrazed quadrats the five most dominant species represented 48.09% and 61.18% of dominance, respectively. These included graminoid species in all quadrats combined and ungrazed plots. *Rumex nepalensis* and *Primula dendiculata* were the only forbs representing the five most dominant species in grazed plots. The other three species were represented by graminoid species namely *Kobresia nepalensis*, *Festuca coelestis* (Poaceae) and *Juncus sp* (Juncaceae).

## ANPP

The mean ANPP in all quadrats combined was 756.46 g/m$^2$ (Table 1). Productivity showed a decreasing trend with elevation with the maximum (829.77 g/m$^2$) in lower zones and minimum (674.93 g/m$^2$) in the higher zone quadrats (Fig 2E, Table 1). The analysis showed that elevation had a significant (p<0.001) role on ANPP. The mean ANPP was highest in July for all grazed quadrats in lower zone, middle zone and all zones combined but it was highest in August in higher zone (Fig 2K).

Productivity was significantly higher in grazed areas than in ungrazed areas (Table 1). Overall, the average ANPP in grazed areas was 32.07% higher than in ungrazed areas. Maximum ANPP (955.27 g/m$^2$) was found in quadrats laid in the grazed areas in the lower zone and minimum (595.9 g/m$^2$) in ungrazed areas in the higher zone (Table 1). Grazing significantly increased ANPP by 35.64%, 33.28% and 26.52% in the lower, middle and higher zones, respectively. The analysis showed that grazing had a statistically significant effect (p<0.001) on productivity within each elevation zone and all elevation zones combined (Table 1). The linear regression revealed a statistically significant (p<0.001, R$^2$ = 0.457) relationship between species richness and ANPP (Fig 2L).

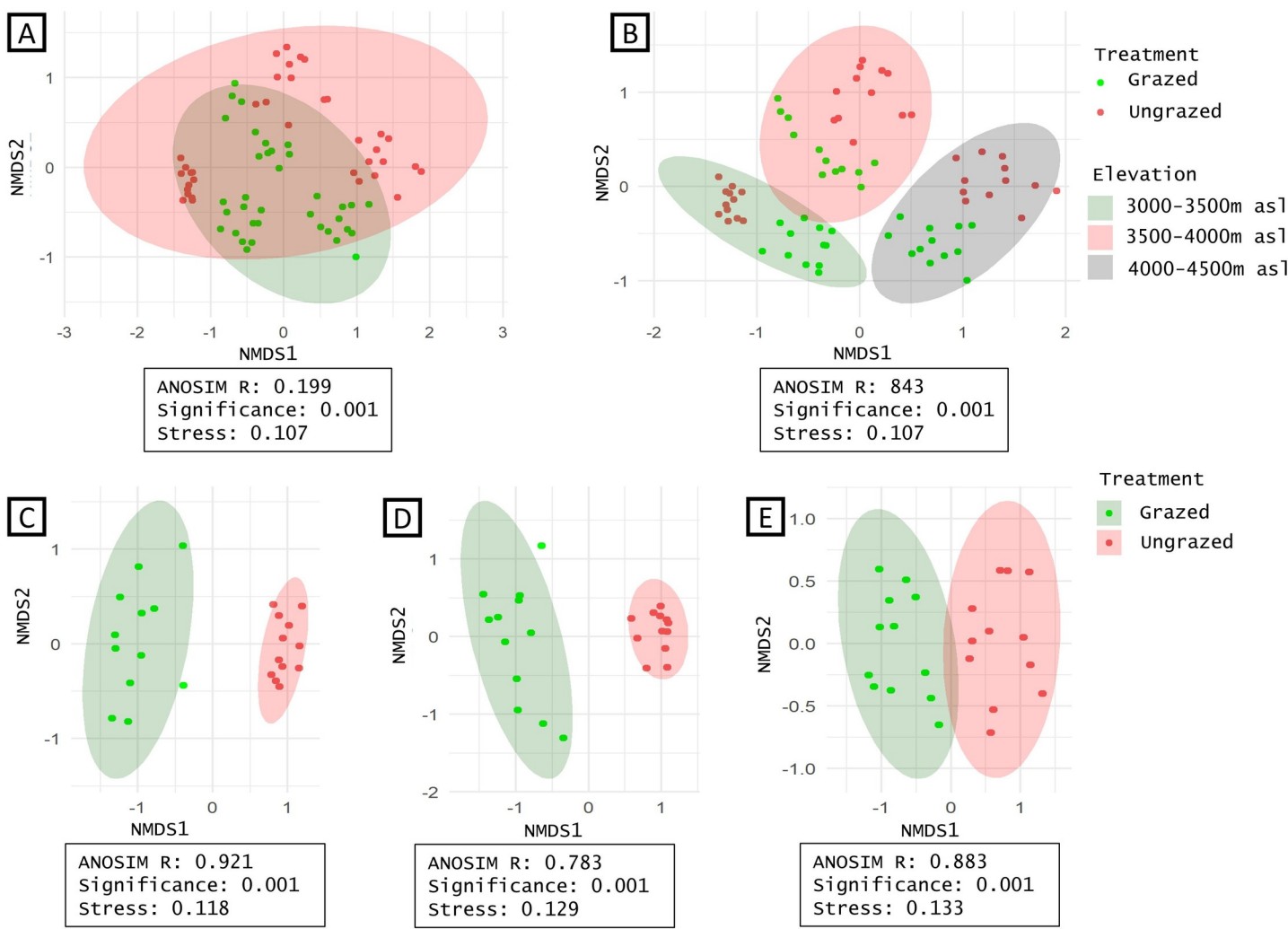

**Fig 3. Non-metric Multidimensional Scaling (NMDS) based on Bray Curtis similarity matrix of plant communities.** Analysis of similarity (ANOSIM) statistic R and its significance are indicated in boxes. Groups are represented by ellipses (95% CI around centroids). NMDS ordination of 153 species grouped according to grazing treatment (A) and elevation gradient (B); NMDS ordination of species in lower (C), middle (D), and higher elevation (E) zones respectively grouped according to grazing treatment.

## Discussion

### Impact of grazing on species diversity and ANPP

Studies from around the world show varied effects of grazing on vegetation [2,21,22,64–66]. A meta-analysis of sixty-one studies from the Tibetan plateau showed grazing mediated increase in species diversity but a decrease in above ground biomass [67]. In the Himalaya grazing has been shown to increase species richness under moderate grazing intensity [38,68,69]. The results of this study show an increase in species diversity indices in grazed areas. Species richness in grazed areas increased around 90% in all elevations combined and doubled in the middle and lower zone. In the higher zone species richness increased by 61.5%. Studies from the region show mixed results with respect to the impact of grazing on ANPP [38,69–71]. While Kala and Rawat's [70] study in the Western Himalaya demonstrated a decrease in ANPP by 31% under grazing treatment, Bagchi and Ritchie [69] showed an increase of ANPP in the

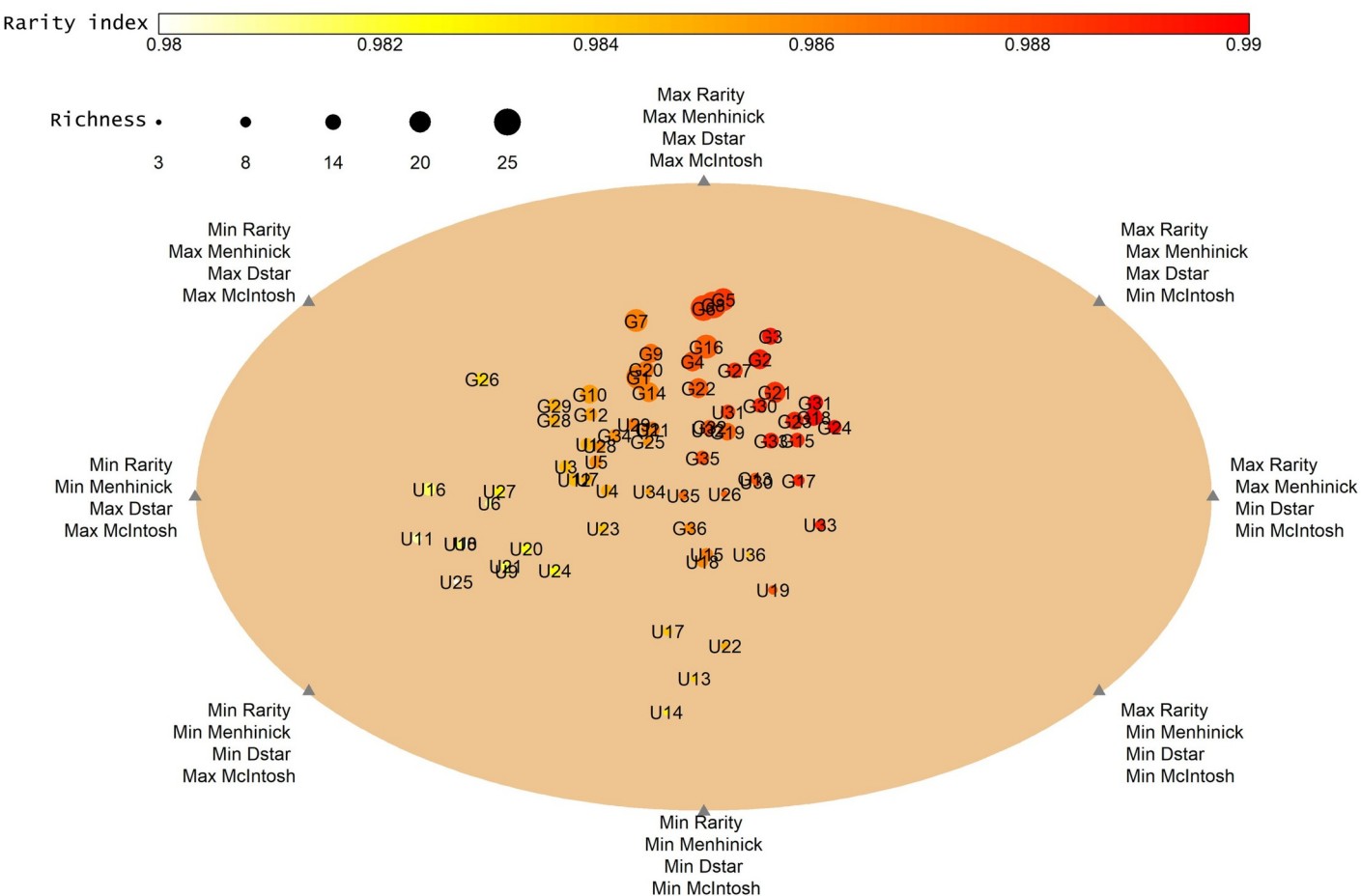

**Fig 4. Spatial distribution of 72 quadrates based on four diversity indices (rarity- Leroy's rarity index, Menhinick -heterogeneity, Dstar-taxonomic distinctness and McIntosh index-evenness).** The size of the circles increases with richness, and a color-code (gradient) is used to indicate the rarity index. Circles are labelled G for grazed quadrates and U for ungrazed quadrates. Numbers in circles represent quadrate numbers (1–12 lower zone, 13–24 middle zone, 25–36 higher zone).

Trans-Himalaya by 61%. They attributed this increase to preferential investment in above-ground fraction. The results are consistent with herbivore-mediated positive feedback effect on ANPP and nutrient availability between soil nutrients [72,73]. In my study ANPP increased by 32.07%, 35.64%, 33.28% and 26.53% in all elevations combined, lower, middle, and higher zones respectively under grazing treatment.

The results of this study demonstrate the importance of the traditional grazing system in maintaining high plant diversity and productivity. Significantly higher diversity indices and productivity in grazed areas suggest plant community composition, structure and function benefit or depend indirectly on grazing for their persistence in the high-elevation alpine pastures of the Himalaya. Grazing may be a driver for higher species diversity in three ways.

First, in free grazing systems like in the study site, higher diversity could be facilitated by the important role livestock play as seed dispersers. Seed dispersal by grazing animals is important in grasslands throughout the world, being vital for conservation of not only species but genetic diversity [74–76]. Grazing animals are dispersal vectors through epizoochory (transporting outside the body e.g. on fur) or endozoochory (transporting after ingestion). Studies have found higher survival of small seeds in the digestive tract of mammals increasing effectiveness of endozoochory [77–79]. This could explain higher PD of small seeded species such

as Orchidaceae and Asteraceae (especially *Anaphalis lactea* and *Lentopodium leontopodioides)* in grazed plots than ungrazed plots.

Second, grazing results in the reduction in abundance of dominant species by selectively removing palatable species [67]. This explains higher PD for graminoid species (that are palatable) in ungrazed areas. The results show that graminoid species represent 78.19% of the total PD in ungrazed quadrats as compared to 44.34% in grazed quadrats. Graminoid species of Cyperaceae and Poaceae families were more palatable than Juncaceae which was grazed after most palatable species were eaten. Additionally, higher PD of less palatable families such as Asteraceace, Primulaceae and Rosaceae (dominated by *Potentilla* spp) was found in grazed plots.

Third, grazing can increase spatial heterogeneity thus increasing habitat diversity. Free ranging livestock may modify soil properties directly by recycling nutrients and concentrating it from a larger area into small patches through their dung and urine deposition [64,80–82]. Grazing may also affect soil properties indirectly through their forage preference and grazing intensity. This may selectively affect the survival of some plants that affect soil properties through their rhizosphere [83,84].

Grazing mediated dispersal, foraging preferences and spatial heterogeneity would increase niche breadth and overlap and may promote patches of rarer species thus increasing diversity.

A strong positive correlation between productivity and rainfall has been well documented across the globe [64,85–87]. In the study site monthly productivity in grazed plots were closely associated with rainfall. The monsoon season in the region starts in May, peaking in July and gradually ending in September and October. Peak ANPP in lower and middle elevation quadrats was reached in July and gradually tapered off with the receding monsoon winds (Fig 2K). The monsoons gradually travels northwards, towards the higher elevation explaining why the higher elevation quadrats showed a slight delay in peak ANPP (in August) (Fig 2K). Interestingly the migration pattern of the local communities is closely associated to temporal trends in productivity with the Lachenpas moving up to the lower and middle zones in June and July and the Dokpas moving down to the lower elevations (study design's higher zones) around mid-July. Higher ANPP in grazed plots suggests grazing mediated species diversity is an important driver of ANPP.

## Relationship between species diversity and ANPP

A strong positive correlation between diversity and productivity (Fig 2L) suggests higher biodiversity is an important factor in driving ecosystem functions. The results demonstrating that biomass production increased with diversity conforms to numerous studies from other parts of the world [88–91]. There are three explanations proposed based on experimental studies to understand the role of diversity on productivity. Firstly, complementarity effects occur when more diverse communities facilitate greater niche partitioning allowing for a more complete utilization of resources resulting in increased biomass production [88,90,92]. Secondly, facilitative effects of some species promote the establishment and survival of other co-occurring species influencing biomass production [89,93]. Thirdly, sampling or selection effects suggesting that more diverse plots have a higher probability of containing species with high biomass [89,91,94,95]. The three effects are not mutually exclusive, rather a combined influence of complementarity, facilitative and selection effect of biodiversity influence biomass in a community.

Studies on the impact of grazing on ecosystems have shown mixed results. Negative effects of grazing have been attributed to high grazing intensity owing to overgrazing and high stocking rates or to low grazing intensity including the abandonment of grazing [2,21,22,64,65,96]. The general consensus is that a light to medium grazing intensity is most beneficial [2,64,66].

This study demonstrates that the pastures in Lachen valley have benefitted from grazing of various livestock suggesting the stocking rates are within or below "carrying capacity". The traditional rotational grazing system practiced by the communities allow for pasture regrowth and recovery, facilitates resilience and limits degradation. Similar results have been shown by studies from other regions underlining the benefits of rotational grazing [97–100].

## Socio ecological systems

Driven by external constraints and the benefits of urbanization, the local communities have diversified toward other sectors, particularly the tourism industry and as porters for the Indian army. The most immediate reason for this change was the complete closure of the international border after the war of 1962 between India and China. This not only ended trade with Tibet, but also led to a loss of access to vast areas of pastures in TAR. In addition to this, the establishment of large army encampments and areas for land mines further reduced areas for grazing to an estimated 40% of the original area [101]. Previously, the transboundary migration also ensured the interaction of a larger pool of pastoral people resulting in intermarriages from a larger region as well as more opportunities to cross breed livestock. Today, most Dokpa children are sent to schools in the city and few return to the rigors of a pastoralist's life. Some Dokpas prefer working as porters and laborers for the Indian army stationed in the region or in the tourism industry as guides, porters, and taxi drivers. Lachenpas too are investing more into the growing tourism industry and less into animal husbandry that show diminishing monetary returns. The loss of grazing pastures and access to trade, reduced opportunities for marriage and cross-breeding livestock, and the exposure to modern education and urbanization have resulted in a reduction in Dokpa population living in the region from 30 households (130 persons) in the 1950's [101,102] to just sixteen families [102]. Fewer Dokpas and Lachenpas investing in pastoralism would eventually result in lesser livestock numbers.

## Conclusion

The paper demonstrates that grazing enhances species diversity and productivity indicating that the pastures in the region have benefited from grazing by livestock. The results are significant in two respects. First, it suggests that a complete ban on grazing, a common practice in protected areas of developing countries [103–105] may not meet its primary objective of conserving biodiversity. Second, the socioeconomic losses because of the grazing ban can be reduced by reintroducing grazing after sufficient research on altered herd sizes and the impacts of specific livestock and grazing intensities. The results suggest that a management strategy that is mutually beneficial for conservation and livelihoods is not just possible but desirable to maintain higher local and regional biodiversity.

Moreover, crucial role of the traditional grazing system on biodiversity and ecosystem function, coupled with disinvestment in pastoralism by the locals poses an imminent threat to the landscape that is an intricate socio-ecological system with a long history of co-adaptation with ruminant animal. Pastoralism is associated with other spillover effects such as illegal timber and fuelwood collection, medicinal plants (often rare and endemic) collection and the negative effects of overgrazing as enumerated by Bhagwat et al. [41]. At the same time this study shows, the absence of grazing may result in loss of biodiversity and reduced ecosystem function. The benefits of the traditional free ranging pastoralism cannot be ignored, and a holistic approach is needed to conserve the rich biodiversity of the region. As studies from the region demonstrate the grazing ban has resulted in reduced incomes [41], increase in monocultures in lowlands, decreased manure production in a state that exclusively practices organic farming [39], spread of gregarious species and perceived increase in human wildlife conflict [41,43].

I conclude with research and policy recommendations that should be part of an improved grazing management policy in the state.

1. Experimental studies reintroducing grazing in small areas in regions where the grazing ban was enacted need to be conducted. Studies with the aim to understand the role of different livestock species and different grazing intensities on biodiversity structure and function need to be conducted.

2. Climate change is among the biggest threats to Himalayan rangelands, a region of immense significance for ecosystem services provided to atleast one fifth of humanity and among the most biodiverse in the world [34,37]. Further research on the role of grazing on combating shrub encroachment in response to climate change that threatens habitat loss is imperative.

3. The present grazing ban policy needs to be revisited with a more holistic approach

   a. To understand the role of altered herd sizes on diversity and ecosystem function. A complete ban of grazers meets neither conservation nor livelihood goals. A comprehensive approach taking into account social, ecological, and economic strategies would be required to meet sustainable goals.

   b. Explore a payment for ecosystem services schemes for the services provided by pastoralists in maintaining biodiversity and ecosystem function. These schemes should be aimed at pastoralists moving away from practicing traditional pastoralism. Similar schemes have been used in Switzerland where "*general direct payments*" are made for upland farmers to encourage grazing livestock in more difficult conditions [106].

   c. Explore the link between the grazing ban policy and increasing incidences of human wildlife conflict as reported by Bhagwat et al. [41]. An efficient compensation scheme for human wildlife conflict that is affecting marginalized communities needs to be developed.

   d. Enact reforms to improve livelihoods of ex-herders who are products of the grazing ban policy. While the government has made efforts to reduce this economic impact yet Gupta et al. [39] report frustration among ex-herders in response to "*a lack of reform to improve ex-herders' livelihoods*" post ban.

## Supporting information

**S1 Table. List of species with details on their presence (+) or absence (-) in grazed and ungrazed plots sampled.** Unidentified taxa were those that we could not identify at a species, genus, or family. From this preliminary list, a refined list was prepared where synonyms were combined under a single epithet, based on the taxonomic literature. Where genus or species was not identified we used a number. For the purpose of the analysis, name of the species was not imperative but making sure they were distinguishable species.
(DOCX)

## Acknowledgments

I would like to thank Dr. Kamaljit Bawa for offering vital comments and edits and Dr. Mahesh Sankaran for his advice on the experimental design. Dharmendra Lamsal assisted with parts of the data collection. I would like to thank the Lachenpa and Dokpa communities of Lachen valley and the Dzumsa for their support and enthusiasm. I acknowledge the Department of Forest, Environment & Wildlife Management, Government of Sikkim for permissions to conduct research in the remote Himalayan localities.

## Author Contributions

**Conceptualization:** Tenzing Ingty.

**Data curation:** Tenzing Ingty.

**Formal analysis:** Tenzing Ingty.

**Funding acquisition:** Tenzing Ingty.

**Investigation:** Tenzing Ingty.

**Methodology:** Tenzing Ingty.

**Project administration:** Tenzing Ingty.

**Resources:** Tenzing Ingty.

**Validation:** Tenzing Ingty.

**Visualization:** Tenzing Ingty.

**Writing – original draft:** Tenzing Ingty.

**Writing – review & editing:** Tenzing Ingty.

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
