## [Decision Letter · Decision Letter 0]

16 Sep 2020

PONE-D-20-24843

Pastoralism in the highest peaks: Role of the traditional grazing systems in maintaining biodiversity and ecosystem function in the alpine Himalaya.

PLOS ONE

Dear Dr. Ingty,

Thank you for submitting your manuscript to PLOS ONE. After careful consideration, we feel that it has merit but does not fully meet PLOS ONE’s publication criteria as it currently stands. Therefore, we invite you to submit a revised version of the manuscript that addresses the points raised during the review process.

We look forward to receiving your revised manuscript.

Kind regards,

Bhoj Kumar Acharya, PhD

Academic Editor

PLOS ONE

Additional Editor Comments:

The MS addresses an important issue but there are problems associated with study design, data analysis and flow. Kindly go through the comments provided by all the three reviewers. Additionally, I have provided editorial comments in the pdf file (yellow highlighted with comments in the sticky notes). Address the comments of the reviewers and editor and resubmit the MS.

Journal Requirements:

2. We note that Figure 1 in your submission contain map images which may be copyrighted. All PLOS content is published under the Creative Commons Attribution License (CC BY 4.0), which means that the manuscript, images, and Supporting Information files will be freely available online, and any third party is permitted to access, download, copy, distribute, and use these materials in any way, even commercially, with proper attribution. For these reasons, we cannot publish previously copyrighted maps or satellite images created using proprietary data, such as Google software (Google Maps, Street View, and Earth). For more information, see our copyright guidelines: http://journals.plos.org/plosone/s/licenses-and-copyright.

2.1.    You may seek permission from the original copyright holder of Figure 1 to publish the content specifically under the CC BY 4.0 license. 

2.2.    If you are unable to obtain permission from the original copyright holder to publish these figures under the CC BY 4.0 license or if the copyright holder’s requirements are incompatible with the CC BY 4.0 license, please either i) remove the figure or ii) supply a replacement figure that complies with the CC BY 4.0 license. Please check copyright information on all replacement figures and update the figure caption with source information. If applicable, please specify in the figure caption text when a figure is similar but not identical to the original image and is therefore for illustrative purposes only.

Reviewers' comments:

Reviewer's Responses to Questions

**Comments to the Author**

1. Is the manuscript technically sound, and do the data support the conclusions?

Reviewer #1: Yes

Reviewer #2: Yes

Reviewer #3: Partly

2. Has the statistical analysis been performed appropriately and rigorously? 

Reviewer #1: Yes

Reviewer #2: Yes

Reviewer #3: No

3. Have the authors made all data underlying the findings in their manuscript fully available?

Reviewer #1: Yes

Reviewer #2: Yes

Reviewer #3: No

4. Is the manuscript presented in an intelligible fashion and written in standard English?

Reviewer #1: Yes

Reviewer #2: No

Reviewer #3: Yes

5. Review Comments to the Author

Reviewer #1: This study is very interesting to scientifically validate to show the impact of grazing exclusion policy of the Department of Forest and Environment, Government of Sikkim. Grazing in the forests starting from the Sub-tropical, temperate, alpine forests and the Trans Himalayan rangelands was and is a traditional system of animal farming in the Sikkim Himalaya. During 1998-2005 the Department of Forest and Environment, Government of Sikkim imposed “Ban on Grazing” in Protected Areas and Reserved Forests in the sub-tropical, temperate and alpine areas of East, South, West and North Sikkim with the objective of conserving the biodiversity and improvement of degraded forest. Grazing was considered as detrimental to forest, and biodiversity, illegal activities and extraction. While the scientific validation of the impact of grazing on forest and biodiversity deterioration was not available with the policy implementing agencies.

Therefore, although carried out only in a specific location in North Sikkim, this study has provided scientific knowledge and understanding that grazing exclusion will result into loss of species richness and decline ecosystem functioning. Thus, this study can be a basis to the policy implementing agencies for revisiting the “Grazing Ban Policy” and develop a new policy framework. Equally, this study will also provide opportunities for the researcher, scientists and planners to take forward the similar study in other parts of Sikkim in particular or the Himalaya at large.

Specific comments

Abstract:

In the abstract you mentioned “This paper explores the impact of the traditional pastoral system on high elevation plant species in Lachen valley, one of the few regions of Sikkim where the grazing ban was not implemented.”

Clearly mention in the abstract which were the current grazing stands and which were the ungrazed forests or plots or treatments you selected in your study. You have not mentioned this and thus following your previous sentence of the study site in the Lachen Valley your sentence speaking about your result does not match in the subsequent sentences.

Please give a clear recommendation as to what the policy makers should do. Do they need to revisit their policy, develop strategic short-term and long term strategic action plans through government notification and initiate regulated grazing allowing the herders to graze livestock??

Introduction

The last paragraph should go to the discussion section.

1. The author can develop a GIS based Hyposometric map with the location of the sampled elevations map along the elevation 3000-4500 m with drainage systems, name of the study sites etc. in a DEM.

2. How did you differentiate the grazing with the domestic ungulates and the wild ungulates ?

3. When and where was the fencing laid, was that laid in all the study sites ?

4. What was the study period of this research ?

5. Which were the livestock, domestic as well as wild ungulates, season of grazing, and the season of your study, is not clear.

6. While the grazed stands were found to be with higher species richness in all the elevation study sites, please give a list of common palatable and unpalatable species found in both the grazed and unglazed sites. This will help policy maker to understand the conservation of important, or the threatened species.

7. Please give number of Dokpa households under Lachen Dzumsa (Gurudongmar-Tsholhamu, and Muguthang), as you have said the population is declining. You have mentioned in the last paragraph before the conclusion section that over a dozen Dokpas are there (???).

8. It is important to note the stocking density to identify the impact of grazing, please justify your result by showing the stocking density in the study elevations. Did you come across any location of overstocking ??

9. Which species were dominant in the three study elevations, as you said the dominant species were reduced due to grazing. Were they all palatable ?

10. Any information on Pray-Predator relationship, impact of grazing on wildlife, livestock depredation etc. will add value to the MS.

11. There is a notion that the Human-Wildlife conflict has increased after ban on grazing, please justify if any such information could be generated during the course of this study.

12. What was the species composition in the grazed plots, how did you identify the species as the palatable species are already grazed by animals ?

13. Clearly mention in your result the species composition based on their season of appearances, snowfall period

14. It would be nice to give high resolution pictures of the stand of both grazed and un-grazed stands.

15. What kind anthro0pogenic pressure were there in grazing stands other than grazing ?

16. Also show in your result what is the species richness in the three elevations ?

17. How did you measure ANPP in the grazed and ungrzaed plots, as the plant biomass of grazed plots were eaten up by animals ?

18. Which species do you think are dispersed by animals ?

Reviewer #2: General Comments:

1. Author should give special attention in the spellings of the botanical names of the plant species, grammar and punctuation

2. Repetition of words encountered in places which needs to be checked

Specific Comments:

1. Abstract

• Such long introduction in abstract is not required. Just restrict to one or two sentences.

• Point out the methods applied for the study in one or two sentences

• Highlight the most significant findings in the result section in one or two sentences instead of generalizing it. For eg., The multidimensional scaling and ANOSIM (Analysis of Similarities) pointed to significant differences in plant species assemblages in grazing and ungrazed areas. Significant at what level?

2. Introduction

• The author have mentioned in third paragraph (line 6 – 8) that “Open grazing was phased out over the next six years in all reserved forests with exception of Lachen and Lachung valley and alpine pastures in the Kangchenzonga National Park”. Logically, grazing is strictly not allowed in National Parks. This statement may negatively impact the UNESCO’s World Heritage Site status of the KNP. Author should avoid using such statement or should provide supporting references.

• The last paragraph in the introduction section looks more like a part of the discussion based on the present study. Hence, move the paragraph to the discussion section.

3. Materials and Methods

• Normally, above 2000 m asl elevation are considered as higher elevation. Therefore, it would not be appropriate to label the elevation above 3000 m asl as lower elevation zone, middle elevation zone or high elevation zone. Try to find some suitable terminology or replace it with lower zone , middle zone and upper zone duly mentioning the elevation range in the text as well as in the map.

• Method is vague. How is it possible to have both grazed and un-grazed area in the same plot of 30 x 30 m? Needs to be clarified. Further, the unit must be accurate. Either write 30 m x 30 m or replace it with 30 x 30 sq. m.

• Author have mentioned the establishment of 12 movable enclosure cages to study the biomass growth. It is not clear weather all the cages were set up in the same plot at a time. If not then, it is important to mention as how many cages were set up in each plot.

• Major rewriting of the methods under the sub-heading “Impact of grazing…..”, a part of the vegetation sampling section is needed to clear the confusion.

• Under the subheading 1, 2 para, line 4: The standing biomass of was……..what does it mean? please specify or correct the statement.

• Data analysis: Table1 & 2 is the part of the result section; hence, remove the mention of table 1 & 2 from the data analysis section.

• What is the logic behind calculating Shannon Index and Simpson diversity index? As both the indexes calculates the diversity taking into account the species richness and the relative abundance of the species. It is better if the author calculates the dominance index.

4. Results

Species diversity indices:

• Why the same result is depicted in table as well as the box plot (Fig 2 A-D, Table 1). Either use table or figure.

• Para 1, line no.: 8 & 9. Author have mentioned that “Grazing had a significant (ANOVA p<0.001) effect all diversity indices in each elevation zone and all elevation zones combined but from table 2, it is clear that the effect of grazing is not significant for diversity indices in higher elevation. Address this mistake.

• Para 1, last 3 lines. Rewrite the sentence “Unlike grazing, elevation did not have a significant effect (ANOVA p<0.05)diversity indices (fig. 2G-I, table 2) except S (fig. 2F, table 2)” as “Unlike grazing, elevation did not have a significant effect on diversity indices (Fig 2G-I) except S (p<0.05) [Fig. 2F, table 2)”.

• Para 2. Line no. 2-3. The sentence is unclear whether the author is trying to compare the species composition between grazed and un-grazed treatment in different zones or amongst the replicates of grazing treatment in different zones. Needs rewriting.

• ANPP: Rewrite the sentence “The mean ANPP for all grazed quadrates (all elevation zones combined), lower and middle elevation zone grazed quadrates was highest in July (Fig 2K) and in the higher elevation zone grazed quadrates in August (Fig 2K)” as “The mean ANPP was highest in July for all grazed quadrates in lower zone, middle zone and all zones combined but it was highest in August in higher zone (Fig 2 K)”.

5. Discussion

• It should focus more on discussing the result of the present study duly supported by the relevant studies in other Himalayan region. Needs major rewriting especially the first two parts.

Reviewer #3: The paper highlights an interesting phenomenon of imposing a ban on grazing in high altitude pastures. Such bans were imposed with an understanding that removing cattle/other grazers from pastures will help reinstate the diversity of vegetation communities in alpine meadows. However, recently more studies globally are finding that there is a fine line between overgrazing and no grazing, to maintaining diversity in pastures. The author has researched an area, both in location and concept, which is important, especially in light of the changing climate and the associated management policies influencing pastoralism.

The manuscript has been written well and the research design appears to be rigorous. However, the description I believe is a little confusing. The way I understood the design is:

Elevation: Low, Middle, High

Blocks: G (grazed) and U (ungrazed)

Replicates: 3/elevation [3x3 = 9]

Plots: 30x30m, one per block [total = 9] This is confusing – was the 30x30m plot equally straddling G and U areas?

Sampling quadrats: 24 per block (12G and 12U) [24, 1x1m quadrats/elevation, and a total of 216 for the study – while implied (perhaps), this should be mentioned in the text]

If this is correct, a simple chronological statement or a schematic of the levels and arrangement will make following the experimental set-up easy for readers.

One of my major concerns with the manuscript is the analyses carried out with the diversity values. Here the author has used a variety of diversity measures – Shannon entropy, Simpson concentration, and Pielous’s evenness. Nothing is alarming about these indices by themselves, but the problem arises once these values are further analyzed, say using an ANOVA. First off, these indices are limited in range – usually between 0 and 1, for Shannon’s entropy when normalized, for example; exhibiting a floor-ceiling effect and violating the assumptions of a traditional ANOVA. This often leads to wrong results and comparisons. Details of this problem may be further understood by reviewing the paper by Lou Jost in Oikos 2006 – Diversity and Entropy. There are methods suggested in this paper that makes comparing entropies more meaningful. Therefore, instead of using an ANOVA to tease apart the difference among treatments, I suggest that the author use a more comprehensive and inclusive algorithm such as the DER – for comparing species diversity. See Guisande et al. 2017. This can be done in R (package EcoIndR). These are more contemporary techniques and overcome the flaws of other traditional comparison techniques.

Please see attached an annotated content extracted pdf (lit rev not included) of the manuscript for minor edits that will help the paper read well.

Other than that, the research represented by the manuscript is timely and important in understanding ways of restoring the plant diversity of the alpine regions of the Himalaya and elsewhere, especially using adaptive management policies.

6. PLOS authors have the option to publish the peer review history of their article (what does this mean?). If published, this will include your full peer review and any attached files.

Reviewer #1: **Yes: **DR GHANASHYAM SHARMA

Reviewer #2: No

Reviewer #3: No

---

## [Author Response · Author response to Decision Letter 0]

4 Nov 2020

1. Editors comment:

Check the spelling

Response

Spelling of Rhododendron barbatum corrected

2. Editors comment:

repetition

Response

Corrected throughout the paper

3. Editors comment:

Ecosystem function?

In the paper I use above ground net primary productivity (ANPP) an integrative measure of ecosystem function. This is a well-established practice where ANPP is used as a proxy for ecosystem function.

 I had mentioned it in the abstract and have added it to the 2 broad questions in the Introduction section with citations.

4. Editors comment:

This part is more appropriate for conclusion rather than Introduction. Place it appropriately

Response

I have made the edit and placed it in the conclusion section

5. Editors comment:

Does the notification says so? It is better if you cite the notification stating the relaxation on grazing ban given to Lachen. Can you mention the rationale why Lachen and Lachung were excluded?

Response

I have added citations for two papers discussing grazing being allowed in selected areas of KNP and one citation for grazing in Lachen and Lachung.

I have not seen the exact notification and do not know why Lachen and Lachung have been excluded.

6. Editors comment:

Do we have more recent data? 

Response

This is the most recent I found. I am assuming the livestock census like the general census is published every decade. 

7. Editors comment:

But these areas also have effect on species diversity due to hay extraction.

Response

Yes. That is why I explicitly define what I mean by ungrazed areas. These are “areas with a history of no grazing for ateast 10 years with permanent fences around them. These areas are used by the locals to make hay.”

8. Editors comment:

Remove the word elevation and retain lower zone, middle zone and upper zone. Follow it throughout the text

Response

Changes made throughout the manuscript.

9. Editors comment:

How big were the blocks? Were they large enough to have sufficient distance between the grazed and ungrazed areas?

I have made changes to the Vegetation sampling subsection to clarify the experimental design. There were six 30mX30m plots. This included 3 in grazed areas (one for each elevation zone) and 3 in ungrazed areas (one for each elevation zone). In each plot twelve 1mX1m quadrats were laid to quantify diversity. Thus, for each elevation zone I had 24 quadrats (12 in grazed and 12 in ungrazed) with a total of 72 quadrats in all elevations and treatments where I collected data for species diversity. This is explained in the edited lines as 

“I used a randomized block design with two treatments (grazing and ungrazed) and three replicates (lower zone =3000-3500m; middle zone =3500-400m; and higher zone= 4000-4500m) making a total of six 30mX 30m plots (2 treatments X 3 replicates)”

I made further corrections in the subsection- Impacts of grazing on species diversity and above ground net primary productivity (ANPP)

“Diversity: In each 30mX30 m plot 12 quadrats (1mX1 m) were randomly laid. We had 36 quadrates in grazed areas (12X 3elevation zones) and 36 in ungrazed areas. Thus, a total of 72 quadrats were laid (24 in each zone/ replicate). Data on species composition and percentage cover was collected from each quadrat for herb species only (forbs and graminoids).”

10. Editors comment:

Within this small area of 30m x 30 m, how it is possible to have grazed and ungrazed areas. It is not convincing because grazing animals are not restricted within 15 m (roughly as per your design) are for grazing

Response

Please see response comment above. The grazed areas were infact much larger than 30mX30m but laid our 12 plots within that plot (30mX30m) only. 

11. Editors comment:

You estimated only herbs or woody vegetation were also included in the quadrat? If woody vegetation were included, your quadrat size is too small and if they are not included, how did you avoid them because in pastures also there are few woody species sparsely grown?

Response

Only herbs (forbs and graminoids) species were sampled in each quadrat. I have added this point in the subsection- Impacts of grazing on species diversity and above ground net primary productivity (ANPP). The grazing pastures we sampled (Phemkaru, Yathang and Chopta valley) had a negligible number of woody vegetation and none fell in our quadrats. In the ungrazed plots (fenced areas) there were few woody vegetation and only along the edges which we anyway avoided to avoid for edge effect.

12. Editors comment:

How did you do it in the ungrazed area?

Response

In the ungrazed areas ANPP was collected at peak standing crop once a year. I have edited the last line in the subsection Impacts of grazing on species diversity and above ground net primary productivity (ANPP) to

“In ungrazed areas twelve 1mX1m quadrats were laid and biomass was collected at peak standing crop (annually).”

13. Editors comment:

Was there a facility to do this in Lachen??

Response

No, there was no facility to oven dry in Lachen. The biomass collected was stored with blotting paper and was sent occasionally (about every 1 week to Gangtok where we had an oven to dry it).

14. Editors comment:

Please see the comments of Reviewer 3 and modify the analysis based on the comments

Response

I have conducted the analysis suggested by reviewer 3. I used the DER algorithm comparing the differences between assemblages in in terms of four components of diversity (1) rarity (Leroy’s rarity index), (2) heterogeneity of the grassland community( Menhinick index), (3)evenness (McIntosh index)and (4) taxonomic distinctness (Dstar). I appreciate reviewer 3’s suggestion and feel the new analysis greatly enhances the paper. I also used the Kruskal-Wallis test to test the difference between grazing treatments and between different elevations zones for diversity indices and ANPP (table1). Table 1. needs no changes since the results (significance categories) are the same. Table 2 has been omitted since the section on ANOVA has been deleted.

15. Editors comment:

Mention the packages used

Response

I have mentioned it at the end of the data analysis subsection. All analyses were performed on R. The NMDS and ANOSIM analysis were performed using the vegan package in R. I corrected my mistake- vegan is a package in R not a function.

I have also mentioned the package EcoIndR for the DER algorithm.

16. Editors comment:

Provide the list of plant species observed during sampling as supplementary material

Response

Added.

17. Editors comment:

Do not start sentence with numbers

Response

Changes made.

18. Editors comment:

In my viewpoint this analysis does not have any meaning. Also see the comments of Reviewer 3.

Response

Please see response to comment 14. In addition to this comparing diversity indices in grazed versus ungrazed plots is a well-established practice as is evidenced from several published papers for many years and I have compared them using a non-parametric Kruskal-Wallis test as suggested by reviewer 3. The results were the same (significance categories) and table one needs no changes.

19. Editors comment:

This analysis is not clear. Did you compare the composition among among quadrates? If so, what is the relevance to the objective of this study?

Response

Yes, the Analysis of similarity (ANOSIM) was used to check the degree of dissimilarity of species compositions. As a stand alone the results of the analysis are very interesting but in conjunction with the analysis on species richness, other diversity indices and ANPP, it does give a clearer idea of the role of grazing in diversity and ecosystem function. While without the ANOSIM the results show higher diversity and ANPP in grazed areas, the ANOSIM indicates grazing mediated difference in species assemblages/compositions may be driving these differences. 

20. Editors comment:

This is an amazing finding. Hope you have highlighted the reason in the discussion section

Response

Please refer to response to comment 28

21. Editors comment:

Fig. 

Number below ten are written in words

Response

Changes made throughout the paper

22. Editors comment:

plant diversity??

Response

Change made

23. Editors comment:

Use elevation uniformly throughout the text. Altitude and elevation are represented in different ways

Response

Changes made

24. Editors comment:

This sentence contradicts with your previous sentence

Response

I can see how the words “less palatable” can be confusing in this point. The species are less palatable but not unpalatable. They are still eaten by livestock unlike the species in the ungrazed plots and the PD values of these species are lower in ungrazed plots. The point I am making is small seeded species that studies show have higher survival during endozoochory have higher PD vlues in grazed versus ungrazed plots. I am not comparing it within grazed plots. 

I have removed the words “less palatable” to remove any confusion. It can be reinserted if you feel otherwise.

25. Editors comment:

This paragraph looks odd here and away from the objectives of the study

Response

The first and second objective both look at the effects of grazing on plant diversity and productivity. This paragraph primarily discusses ANPP. Not acknowledging the role of rainfall in ANPP trends would suggest grazing is the single most important factor and presents an incomplete picture hence this paragraph is important for the discussion. 

26. Editors comment:

But is is nowhere shown in the result section

Response

Thank you for pointing this out. I have a reference to “Fig 2K” in this section. This paragraph discusses Fig 2K and shows how ANPP is different in different elevation zones and follows the general rainfall patterns. The results section does refer to figure 2K stating that the ANPP is highest in July (in lower, middle, zones and all combined) and August (in higher zone). 

While rainfall data is not shown as it was not available for the study site. In the next sentences we do discuss the rainfall pattern for the region. 

27. Editors comment:

…the Dokpas moving down to the higher elevation zones ..

Response

Sentence edited as below to clarify

“….Dokpas moving down to the lower elevations (study design’s higher zones) around mid-July”

28. Editors comment:

I did not see any explanation on why the species composition similarity was very low between grazed and ungrazed areas. It is an interesting findings but need through discussion.

Response

I agree that it is a very interesting result. I do not have an in-depth explanation although the three major points in the discussion, subsection- Impact of grazing on species diversity and ANPP touch upon this finding. The second and third point especially puts forth an argument for this. “grazing results in the reduction in abundance of dominant species by selectively removing palatable species” and “grazing can increase spatial heterogeneity thus increasing habitat diversity”. Higher pressure to reduce dominant species and higher habitat diversity in grazed areas would facilitate species unique to grazed areas. It is also understandable that a plot where no grazing has occurred for over a decade will develop different species compositions. 

Showing differences in species composition is not the objective of the study but the result greatly enhances the discussion on grazing impact on diversity and productivity the primary objective of the study.

29. Editors comment:

How? “This not only ended trade with Tibet, but also led to a loss of vast areas of pastures in TAR.”

Response

I have added the word “access” to read better.

“This not only ended trade with Tibet, but also led to a loss of access to vast areas of pastures in TAR”

Here, I refer to the international border being closed after the war thus the local communities lost access to vast pastures across the border in Tibetan Autonomous region that they used before the border was closed.

30. Editors comment:

What could improve the dairy farming and pastoralism in the region? Highlight some points here

Response

I do not think a step towards improving dairy farming should be taken without sufficient knowledge on the impact that may have. In a region with very dynamic social and economic structures in addition to relatively new pressures due to the burgeoning tourism industry, there are numerous unintended consequences a top down approach may have. Will improved dairy farming incentivize the present and future generations of Dokpas from moving away from nomadic pastoralism? Will a financially successful dairy farming design prevent elite capture and deteriorate the social capital in the present cohorts? These are questions that need to be addressed before investing in improved dairy farming techniques. 

To protect pastoralism, I have 2 points (that are an urgent need) discussed in the conclusion, reiterated below. In addition to this market access and reinstating pride in a traditional livelihood that is often looked at as a last resort by the present generation of local communities would go a long way. 

1. “payment for ecosystem services schemes for the services provided by pastoralists in maintaining biodiversity and ecosystem function”. This has been practiced in other parts of the world notably in Switzerland.

2. “An efficient compensation scheme for human wildlife conflict that is affecting marginalized communities needs to be developed.”

31. Editors comment:

Highlight – in citation

Response

Citation format has been corrected

Reviewer #1: comment 

Abstract:

In the abstract you mentioned “This paper explores the impact of the traditional pastoral system on high elevation plant species in Lachen valley, one of the few regions of Sikkim where the grazing ban was not implemented.” Clearly mention in the abstract which were the current grazing stands and which were the ungrazed forests or plots or treatments you selected in your study. You have not mentioned this and thus following your previous sentence of the study site in the Lachen Valley your sentence speaking about your result does not match in the subsequent sentences.

Response

I have added 2 lines in the abstract clearing this point. The second line clarifies “Ungrazed areas are part of pastures that have been fenced off (preventing grazing) for over a decade and used by the locals for hay formation.”

Reviewer #1: comment

Please give a clear recommendation as to what the policy makers should do. Do they need to revisit their policy….

Response

Conclusion-point 3 explicitly states that the grazing ban policy needs to be revisited. “The present grazing ban policy needs to be revisited with a more holistic approach”

Reviewer #1: comment

….develop strategic short-term and long term strategic action plans through government notification and initiate regulated grazing allowing the herders to graze livestock??

Response

As discussed in the conclusion (point 1) “Experimental studies reintroducing grazing in small areas in regions where the grazing ban was enacted need to be conducted. Studies with the aim to understand the role of different livestock species and different grazing intensities on biodiversity structure and function need to be conducted.” Further studies built on our research would be needed to gather more data on different grazing intensities and different livestock species before initiating “regulated grazing allowing the herders to graze livestock”

Reviewer #1: comment

Introduction

The last paragraph should go to the discussion section.

Response

Thank you for pointing this out. It has been moved to the conclusion section on the recommendation of the editor.

Reviewer #1: comment

1. The author can develop a GIS based Hyposometric map with the location of the sampled elevations map along the elevation 3000-4500 m with drainage systems, name of the study sites etc. in a DEM.

Response

Thank you very much for the recommendation. Figure 1 has been redone using a Digital Elevation Model (DEM) derived from ASTER data

Reviewer #1: comment

2. How did you differentiate the grazing with the domestic ungulates and the wild ungulates ?

Response

As stated in the sub-section under Vegetation sampling. “I studied the combined effects of different grazing species” (wild and domestic).

“It was difficult to separate the effects of different grazers since there was much overlap in the grazing areas shared by the different livestock and wild species. Cows, bulls, goats, horses and a few yak shared the same pastures in the lower elevation zone (3000-3500 m asl). Sheep, yak, cows, bulls, goats, and horses shared the middle elevation zones (3500-4000 m asl) and yak, sheep and a few horses grazed in the higher elevation zone (4000-4500 m asl). Thus, I studied the combined effects of different grazing species.”

Reviewer #1: comment

3. When and where was the fencing laid, was that laid in all the study sites ?

Response

The fenced (ungrazed areas) were laid in all 3 elevation zones. Within Lachen valley there are several areas fenced off by the locals (for atleast 10 years before the study was conducted). These areas are used by the locals to make hay. The herbs are left to grow in these areas throughout the year and removed at the end of the growing season

“Local pastoralists and village representatives were consulted to select grazed and ungrazed areas” One fenced area was in Phemkaru (lower zone), Yathang (middle zone) and Chopta (higher zone). The name of places is mentioned in the edited map (figure 1). The details of fencing is described in the MS in the third sentence of the section on Methods-vegetation sampling 

“The ungrazed areas were defined as areas with a history of no grazing for ateast 10 years with permanent fences around them. These areas are used by the locals to make hay. The herbs are left to grow in these areas throughout the year and removed at the end of the growing season.”

Reviewer #1: comment

4. What was the study period of this research ?

Response

The study was initiated in 2010 but due to the earthquake of 2011 data collection was disrupted. Preliminary data was recollected in 2012 August- October. Finalized data was collected over a three-year period starting from the growing season of 2013 to the end of the growing season in 2015. I have added this to the materials and methods section under the sub-section Vegetation sampling.

Reviewer #1: comment

 5. Which were the livestock, domestic as well as wild ungulates, season of grazing, and the season of your study, is not clear.

Response

Domestic livestock is described in detail in the vegetation sampling section (MS quoted below). We did not monitor wild ungulate population. As stated in this section “I studied the combined effects of different grazing species” this include the domestic livestock mentioned as well as wild ungulates.

“Cows, bulls, goats, horses and a few yak shared the same pastures in the lower elevation zone (3000-3500 m asl). Sheep, yak, cows, bulls, goats, and horses shared the middle elevation zones (3500-4000 m asl) and yak, sheep and a few horses grazed in the higher elevation zone (4000-4500 m asl). Thus, I studied the combined effects of different grazing species. “

Reviewer #1: comment

6. While the grazed stands were found to be with higher species richness in all the elevation study sites, please give a list of common palatable and unpalatable species found in both the grazed and unglazed sites. This will help policy maker to understand the conservation of important, or the threatened species.

Response

Please refer to reply to comment no. 12 

Reviewer #1: comment

7. Please give number of Dokpa households under Lachen Dzumsa (Gurudongmar-Tsholhamu, and Muguthang), as you have said the population is declining. You have mentioned in the last paragraph before the conclusion section that over a dozen Dokpas are there (???).

Response

The number of households has been revised to an older (2002) source with a citation. The number mentioned in the previous version of the manuscript was a result of conversations with Dokpas who mentioned that there were fewer than a dozen families left actively practicing nomadic pastoralism. 

Reviewer #1: comment

8. It is important to note the stocking density to identify the impact of grazing, please justify your result by showing the stocking density in the study elevations. Did you come across any location of overstocking ??

Response

While I agree with the reviewer that calculating stocking density would add value to the study we collected data on type of livestock in plots and not number of individual livestock, and wildlife species were also unaccounted for. If we estimated of stocking densities based on government census data (available for the whole valley and not blocks) it would be inaccurate and misleading to policy makers.

Moreover as Laca in Rangeland Ecol Manage.(2009) point out productivity will be affected by grazing method, bite rates and the defoliation process in addition to stocking density. My study focuses on the vegetation response to traditional grazing system comparing grazed areas versus ungrazed (traditionally fenced areas)

Reviewer #1: comment

9. Which species were dominant in the three study elevations, as you said the dominant species were reduced due to grazing. Were they all palatable ?

Response

Graminoid species were most dominant with Kobresia nepalensis (Cyperaceae) being the most common species in all 3 elevations. Yes, the graminoid species were all palatable.

I have made no changes to the manuscript as this information is present in paragraph 4 of the discussion and paragraph three of the results section.

Reviewer #1: comment

10. Any information on Pray-Predator relationship, impact of grazing on wildlife, livestock depredation etc. will add value to the MS.

Response

I completely agree, unfortunately this was outside the objective and scope of the study and we did not collect information on wildlife.

Reviewer #1: comment

11. There is a notion that the Human-Wildlife conflict has increased after ban on grazing, please justify if any such information could be generated during the course of this study.

Response

I did not come across such information since the study was conducted in areas where grazing was not banned. I have cited papers from parts of Sikkim that have seen a perceived increase in human wildlife conflict after the grazing ban (reference 38[Bhagwat et al.2012], 40 [Tambe et al. 2005], 88 [Gupta et al. 2020])

Reviewer #1: comment

12. What was the species composition in the grazed plots, how did you identify the species as the palatable species are already grazed by animals?

Response

Species were identified as palatable based on literature reviews, consultations with locals and observations. Species were deemed palatable if they met all 3 above criteria. 

I sampled herbs (forbs and graminoid species only). (This has been added in the methods section). Based on the 3 criteria above we were able to confirm that graminoid species were indeed palatable. Especially cyperaceae and poaceae. Juncaceae was less palatable but was still grazed when more palatable species were eaten. (This has been added to the Discussion section- Impacts of grazing on species diversity and ANPP- paragraph 3). I did not classify other species as less palatable unless they did not meet all 3 criteria and grazers avoided these species eg. Anaphalis lactea Lentopodium leontopodioides and Morina longifolia. 

Reviewer #1: comment

13. Clearly mention in your result the species composition based on their season of appearances, snowfall period

Response

Data was collected only during the growing season (mid-April to late October) and not during the snowfall period. I have added as supplementary material a table on species composition.

Reviewer #1: comment

14. It would be nice to give high resolution pictures of the stand of both grazed and un-grazed stands.

Reviewer #1: comment

15. What kind anthro0pogenic pressure were there in grazing stands other than grazing ?

Response

Although close to the road the grazing pastures had no other anthropogenic pressure and maintained solely for grazing. Occasional visits by tourists do occur resulting in trampling only but no construction.

Reviewer #1: comment

16. Also show in your result what is the species richness in the three elevations ?

Response

It is present in table 1. I have added it to the results section too.

Reviewer #1: comment

17. How did you measure ANPP in the grazed and ungrzaed plots, as the plant biomass of grazed plots were eaten up by animals ?

Response

As mentioned in the methods section I followed the difference method by McNaughton et al. (1996) (ref.51). This is a standard method that is most preferred to measure the effect of grazing on plants “since moveable exclosures account for the amount eaten and the amount that regrows between feeding bouts” (McNaughton et al. 1996). Since this is a highly cited paper there is no need to repeat the details of the method more than what is already detailed in the methods section.

Reviewer #1: comment

18. Which species do you think are dispersed by animals?

Response

This is a very interesting question. Any names here would be only conjecture and a separate study focusing on epizoochory and endozoochory would be needed to confirm. Unfortunately, this was outside the objective and scope of the study. If I would have to guess Asteraceae species that may be principally wind pollinated may be occasionally dispersed long distances by endozoochory. 

Reviewer #2: General Comments:

1. Author should give special attention in the spellings of the botanical names of the plant species, grammar and punctuation

Response

Thank you. I have made the edits and corrections

Reviewer #2: comment

2. Repetition of words encountered in places which needs to be checked

Response

Thank you. I have made the edits and corrections

Specific Comments:

Reviewer #2: comment

1. Abstract

• Such long introduction in abstract is not required. Just restrict to one or two sentences.

• Point out the methods applied for the study in one or two sentences

Response

Thank you. Changes made.

Reviewer #2: comment

• Highlight the most significant findings in the result section in one or two sentences instead of generalizing it. For eg., The multidimensional scaling and ANOSIM (Analysis of Similarities) pointed to significant differences in plant species assemblages in grazing and ungrazed areas. Significant at what level?

Response

Thank you. Changes made.

Reviewer #2: comment

2. Introduction

• The author have mentioned in third paragraph (line 6 – 8) that “Open grazing was phased out over the next six years in all reserved forests with exception of Lachen and Lachung valley and alpine pastures in the Kangchenzonga National Park”. Logically, grazing is strictly not allowed in National Parks. This statement may negatively impact the UNESCO’s World Heritage Site status of the KNP. Author should avoid using such statement or should provide supporting references.

Response

I have added citations for two papers discussing grazing being allowed in selected areas of KNP.

Reviewer #2: comment

• The last paragraph in the introduction section looks more like a part of the discussion based on the present study. Hence, move the paragraph to the discussion section.

Response

I have made the edit and placed it in the conclusion section

Reviewer #2: comment

3. Materials and Methods

• Normally, above 2000 m asl elevation are considered as higher elevation. Therefore, it would not be appropriate to label the elevation above 3000 m asl as lower elevation zone, middle elevation zone or high elevation zone. Try to find some suitable terminology or replace it with lower zone , middle zone and upper zone duly mentioning the elevation range in the text as well as in the map.

Response

Thank you for the suggestion, I have made the changes

Reviewer #2: comment

• Method is vague. How is it possible to have both grazed and un-grazed area in the same plot of 30 x 30 m? Needs to be clarified. Further, the unit must be accurate. Either write 30 m x 30 m or replace it with 30 x 30 sq. m.

Response

I have made changes to the Materials and methods section under sub-section vegetation sampling

Reviewer #2: comment

• Author have mentioned the establishment of 12 movable enclosure cages to study the biomass growth. It is not clear weather all the cages were set up in the same plot at a time. If not then, it is important to mention as how many cages were set up in each plot.

Response

I have made changes to the Materials and methods section under sub-section vegetation sampling

Reviewer #2: comment

• Major rewriting of the methods under the sub-heading “Impact of grazing…..”, a part of the vegetation sampling section is needed to clear the confusion.

Response

Thank you. Changes made

Reviewer #2: comment

• Under the subheading 1, 2 para, line 4: The standing biomass of was……..what does it mean? please specify or correct the statement.

Response

I have changed it to “aboveground standing biomass (vegetation)”

Reviewer #2: comment

• Data analysis: Table1 & 2 is the part of the result section; hence, remove the mention of table 1 & 2 from the data analysis section.

Response

Changes made

Reviewer #2: comment

• What is the logic behind calculating Shannon Index and Simpson diversity index? As both the indexes calculates the diversity taking into account the species richness and the relative abundance of the species. It is better if the author calculates the dominance index.

Response

As suggested by reviewer #3 I compared the differences between assemblages in terms of rarity (Leroy’s rarity index), heterogeneity of the grassland community( Menhinick index), evenness (McIntosh index)and taxonomic distinctness (Dstar). The four the components of diversity were combined into a single measurement using the DER_algorithm in the EcoINDR package in R. In addition to this I also calculated proportional dominance. 

If permitted, I would prefer to leave both Simpson diversity and Shannon indices if not I would prefer to leave in the Shannon index. Having both indices may be useful as part of a larger meta-analysis. Recent publications (eg. Lu et al. 2017;Ecosphere) have used both indices as part of their meta-analysis. I think meta-analyses play an important role as a policy tools and my primary data would help feed into larger analyses. 

Reviewer #2: comment

4. Results

Species diversity indices:

• Why the same result is depicted in table as well as the box plot (Fig 2 A-D, Table 1). Either use table or figure.

Response

I am fine with fig 2A-D being removed. Although I would prefer it being retained since the boxplots give an idea of the spread of data that the standard deviations (in the table) fail to display.

Reviewer #2: comment

• Para 1, line no.: 8 & 9. Author have mentioned that “Grazing had a significant (ANOVA p<0.001) effect all diversity indices in each elevation zone and all elevation zones combined but from table 2, it is clear that the effect of grazing is not significant for diversity indices in higher elevation. Address this mistake.

Response

Thank you for catching this. Changes have been made to table 2. 

Reviewer #2: comment

• Para 1, last 3 lines. Rewrite the sentence “Unlike grazing, elevation did not have a significant effect (ANOVA p<0.05)diversity indices (fig. 2G-I, table 2) except S (fig. 2F, table 2)” as “Unlike grazing, elevation did not have a significant effect on diversity indices (Fig 2G-I) except S (p<0.05) [Fig. 2F, table 2)”.

Response

Changes made

Reviewer #2: comment

• Para 2. Line no. 2-3. The sentence is unclear whether the author is trying to compare the species composition between grazed and un-grazed treatment in different zones or amongst the replicates of grazing treatment in different zones. Needs rewriting.

Response

Thank you. I have rewritten the 2 lines

Reviewer #2: comment

• ANPP: Rewrite the sentence “The mean ANPP for all grazed quadrates (all elevation zones combined), lower and middle elevation zone grazed quadrates was highest in July (Fig 2K) and in the higher elevation zone grazed quadrates in August (Fig 2K)” as “The mean ANPP was highest in July for all grazed quadrates in lower zone, middle zone and all zones combined but it was highest in August in higher zone (Fig 2 K)”.

Response

Changes made

Reviewer #2: comment

5. Discussion

• It should focus more on discussing the result of the present study duly supported by the relevant studies in other Himalayan region. Needs major rewriting especially the first two parts.

Reviewer #3: comment

The manuscript has been written well and the research design appears to be rigorous. However, the description I believe is a little confusing. The way I understood the design is:

Elevation: Low, Middle, High

Blocks: G (grazed) and U (ungrazed)

Replicates: 3/elevation [3x3 = 9]

Plots: 30x30m, one per block [total = 9] This is confusing – was the 30x30m plot equally straddling G and U areas?

Sampling quadrats: 24 per block (12G and 12U) [24, 1x1m quadrats/elevation, and a total of 216 for the study – while implied (perhaps), this should be mentioned in the text]

If this is correct, a simple chronological statement or a schematic of the levels and arrangement will make following the experimental set-up easy for readers.

Response

I have made changes to the Materials and methods section under sub-section vegetation sampling that should clarify the study design

Reviewer #3: comment

One of my major concerns with the manuscript is the analyses carried out with the diversity values. Here the author has used a variety of diversity measures – Shannon entropy, Simpson concentration, and Pielous’s evenness. Nothing is alarming about these indices by themselves, but the problem arises once these values are further analyzed, say using an ANOVA. First off, these indices are limited in range – usually between 0 and 1, for Shannon’s entropy when normalized, for example; exhibiting a floor-ceiling effect and violating the assumptions of a traditional ANOVA. This often leads to wrong results and comparisons. Details of this problem may be further understood by reviewing the paper by Lou Jost in Oikos 2006 – Diversity and Entropy. There are methods suggested in this paper that makes comparing entropies more meaningful. Therefore, instead of using an ANOVA to tease apart the difference among treatments, I suggest that the author use a more comprehensive and inclusive algorithm such as the DER – for comparing species diversity. See Guisande et al. 2017. This can be done in R (package EcoIndR). These are more contemporary techniques and overcome the flaws of other traditional comparison techniques.

Please see attached an annotated content extracted pdf (lit rev not included) of the manuscript for minor edits that will help the paper read well.

Other than that, the research represented by the manuscript is timely and important in understanding ways of restoring the plant diversity of the alpine regions of the Himalaya and elsewhere, especially using adaptive management policies.

Response

Thank you very much for the suggestion. I have done the analysis using the DER algorithm comparing the differences between assemblages in in terms of rarity (Leroy’s rarity index), heterogeneity of the grassland community( Menhinick index), evenness (McIntosh index)and taxonomic distinctness (Dstar). I also used the Kruskal-Wallis test in place of the t-test used earlier. I also used the Kruskal-Wallis test to test the difference between grazing treatment and between different elevations zones for diversity indices and ANPP and removed the ANOVA analysis.

---

## [Editor Report · Decision Letter 1]

16 Nov 2020

PONE-D-20-24843R1

Pastoralism in the highest peaks: Role of the traditional grazing systems in maintaining biodiversity and ecosystem function in the alpine Himalaya.

PLOS ONE

Dear Dr. Ingty,

Thank you for submitting your manuscript to PLOS ONE. After careful consideration, we feel that it has merit but does not fully meet PLOS ONE’s publication criteria as it currently stands. Therefore, we invite you to submit a revised version of the manuscript that addresses the points raised during the review process.

We look forward to receiving your revised manuscript.

Kind regards,

Bhoj Kumar Acharya, PhD

Academic Editor

PLOS ONE

**Editor Comments**:

I have carefully evaluated the revised MS along with the response to comments provided by the author. The author successfully revised the MS based on the comments of the Editor and all the three reviewers. Now the MS is in better shape but I still find some minor issues with the MS. Hence, I suggest the author to make necessary correction (as per the suggestion provided in the attached pdf file with yellow mark and sticky note). Additionally, there are lots of typos and grammar here and there. Similarly, uniformity should also be followed. Also check the list of references thoroughly and make them complete. Figure 1 (map of study area) need revisit. I suggest the author to read the MS more critically and make it error free (typos, grammar, uniformity, technicalities, etc). Once done, it may be accepted without external review.

---

## [Author Response · Author response to Decision Letter 1]

21 Dec 2020

1. Please address this comment

Discussion: It should focus more on discussing the result of the present study duly supported by the relevant studies in other Himalayan region. Needs major rewriting especially the first two parts

I have added a paragraph in the discussion section discussing my results and studies from other parts of the world and the Himalaya. 

2. This looks like a phone number rather than a permit number

Thank you for catching this. I have corrected the permit details.

3. Avoid this value in abstract

Value removed

4. Should they be in alphabetical order?

I have added the keywords in order of role/ importance.

5. Delete one dot

Deleted

6. Modify 'after the declaration of grazing ban'

Modified

7. Remove from here and take it to acknowledgments section. The permit number given by you looks like a phone number of the forest department. Provide the permit number

I have fixed the permit number. The journal requires that the permit number is written in the Methods section so I have added it to the Methods paragraph and tried to maintain the flow as below

“The study was conducted in Lachen valley in the North district of the state of Sikkim in India. Necessary field research permits were obtained from the Department of Forest, Environment & Wildlife Management, Government of Sikkim, India – Letter No. 16/SRO/F dated 20/04/2010 and Ref. No. 101 /PCCF/F dated 03/02/2012.”

8. Two 

Has been fixed throughout the MS

9. You have mentioned in your response sheet that you have quoted the reference paper but I do not see them here. To make your argument strong enough, please quote the reference(s).

Citation added

10. Now??

Accepted change

11. Colour code in the legend and map do not match. Please check and rectify

Thank you for catching this. The error has been rectified

12. You are a single author in the article

We changed to I throughout the manuscript

13. excluded would be a better term

Change accepted

14. Check

Deleted extra word

15. Check the framing of the sentence

“The stress function assesses how well the two-dimensional plot represent actual sample relationships calculated” changed to “The stress function assesses how well the calculated sample relationships are represented in the two-dimensional plot”.

16. check sentence framing 

“The analysis showed all diversity indices (S,H,D and J) were significantly higher (p<0.001) in grazed quadrats than ungrazed quadrats…” changed to “The analysis showed that the diversity indices( S,H,D and J) were significantly higher (p<0.001) in grazed quadrats than ungrazed quadrats..”

17. small letter 'l', middle, h(igher), T(able)

Changes accepted.

18. Add “,” delete “.” 

Changes accepted throughout

19. Check whether there is any overlap between Figure 2 and Table 1. I suggest to keep only one, which is most representative to the data

I would prefer to keep both table 1 and figure 2 for the following reasons

a. Figure 2K & L are not represented in table 1. Hence, figure 2K & L is required.

b. Figure 2A to J show the spread of the data while table 1 with standard deviation values shows the variance from means at quadrats level. The boxplots are important to indicate the symmetry (skewness) of the data and median while table 1 shows the mean and standard deviation but cannot indicate the skewness of the data that is informative.

c. Table 1 also quantifies the differences that is important to underline the difference between treatments that if removed will have to be reproduced in the results section.

I have added the units (g/m2) to the ANPP column in table 1.

Based on the above reasons I would prefer to have both table 1 and figure 2 retained in the paper.

20. keep it in bracket

Changes accepted throughout the paper

21. check the sentence framing

“In grazed plots Rumex nepalensis and Primula dendiculata were the only forbs representing the five most dominant species the remaining three were represented by graminoid species Kobresia nepalensis, Festuca coelestis (Poaceae) andJuncus sp (Juncaceae)” changed to 

“Rumex nepalensis and Primula dendiculata were the only forbs representing the five most dominant species in grazed plots. The other three species were represented by graminoid species namely Kobresia nepalensis, Festuca coelestis (Poaceae) andJuncus sp (Juncaceae)”

22. Check

The analysis showed that elevation had a significant (p<0.001) role on ANPP. )

Extra bracket removed

23. check the sentence framing

“quadratsquadrats Productivity was significantly higher in grazed areas than in ungrazed areas (table1).” Corrected to 

“Productivity was significantly higher in grazed areas than in ungrazed areas (table1)”

24. Please remove this part

Removed

25. Repetition

Removed.

26. Do you want to keep this?

 Removed

27. Provide the complete reference

Fixed

28. This is an incomplete reference

Fixed

29. Oikos is an established journal. Just give doi, all these details are not needed.

Link removed

30. Updated version is available

I cited a number from this version that is not available in the updated version.

31. Provide doi rather than this link

Link removed

32. doi

Fixed

33. Colour code in the legend and the map does not match. Correct it

Corrected. Thank you.

---

## [Editor Report · Decision Letter 2]

26 Dec 2020

Pastoralism in the highest peaks: Role of the traditional grazing systems in maintaining biodiversity and ecosystem function in the alpine Himalaya.

PONE-D-20-24843R2

Dear Dr. Ingty,

We’re pleased to inform you that your manuscript has been judged scientifically suitable for publication and will be formally accepted for publication once it meets all outstanding technical requirements.

Kind regards,

Bhoj Kumar Acharya, PhD

Academic Editor

PLOS ONE

---

## [Editor Report · Acceptance letter]

30 Dec 2020

PONE-D-20-24843R2 

Pastoralism in the highest peaks:  Role of the traditional grazing systems in maintaining biodiversity and ecosystem function in the alpine Himalaya. 

Dear Dr. Ingty:

I'm pleased to inform you that your manuscript has been deemed suitable for publication in PLOS ONE. Congratulations! Your manuscript is now with our production department. 

Kind regards, 

on behalf of

Dr. Bhoj Kumar Acharya 

Academic Editor

PLOS ONE